# Cooperative Control of Underwater Vehicle–Manipulator Systems Based on the SDC Method

**DOI:** 10.3390/s22135038

**Published:** 2022-07-04

**Authors:** Aleksey Kabanov, Vadim Kramar, Ivan Lipko, Kirill Dementiev

**Affiliations:** Robotics and Intelligent Control Systems Laboratory, Sevastopol State University, 299053 Sevastopol, Russia; kramarv@mail.ru (V.K.); ivanlipko@yandex.ru (I.L.); iuts@sevsu.ru (K.D.)

**Keywords:** underwater vehicle–manipulator system, cooperative control, state-dependent coefficient, SDDRE-method, mathematical modeling

## Abstract

The paper considers the problem of cooperative control synthesis for a complex of N underwater vehicle–manipulator systems (UVMS) to perform the work of moving a cargo along a given trajectory. Here, we used the approach based on the representation of nonlinear dynamics models in the form of state space with state-dependent coefficients (SDC-form). That allowed us to apply methods of suboptimal control with feedback based on the state-dependent differential Riccati equation (SDDRE) solution at a finite time interval, providing the change in control intensity with the transient effect of the system matrices in SDC form. The paper reveals two approaches to system implementation: a general controller for the whole system and a set of N independent subcontrollers for UVMSs. The results of both approaches are similar; however, for the systems with a small number of manipulators, the common structure is recommended, and for the systems with a large number of manipulators, the approach with independent subcontrollers may be more acceptable. The proposed method of cooperative control was tested on the task of cooperative control for two UVMSs with six-link manipulators Orion 7R. The simulation results are presented in the article and show the effectiveness of the proposed method.

## 1. Introduction

The cooperation of robotic systems has many applications in the industry that require coordinated control, such as repair and welding work, space transportation of objects, and others [1,2]. Most of them do not exclude the possibility of operating in an underwater environment, and this fact significantly complicates the implementation of control methods for each case.

Underwater vehicles are divided into two common type classes [1,2,3,4,5]: remotely operated underwater vehicles (ROVs) and unmanned underwater vehicles (UUVs) [6,7,8,9,10]. The first type of UV requires operator control from an on-surface sea vessel. The second type of UV is equipped with all necessary hardware and can move autonomously.

Autonomous underwater vehicles (AUVs) have been widely used in recent years as an alternative to extremely costly, time-consuming, and risky human underwater operations. Most of the tasks performed by these vehicles are for data-gathering applications. However, there is an increasing interest for their use in inspection, maintenance, and repair operations that require manipulation and interaction with objects in the underwater environment. Nowadays, this is mainly performed by remotely operated vehicles (ROVs) [11] and one of the strongest research trends regards the use of Underwater Vehicle–Manipulator Systems (UVMS) [12,13]. Underwater vehicle–manipulator systems (UVMS) can fulfill underwater sampling, grabbing, operation, and other tasks in addition to observation. It is widely used in marine scientific investigations, marine engineering, and other fields [14].

Most underwater manipulator tasks can be performed more efficiently when multiple manipulators are used cooperatively [15]. On the other hand, the tasks of coordinated control of several underwater robots with manipulators are quite complex, with the main difficulties in implementation related to the communication limitations of the aquatic environment. The use of a single base robot carrier with multiple manipulators overcomes such limitations. This research focuses on the development of coordinated control algorithms for underwater robot manipulators that hold an object and move it along a specified trajectory.

It is important to consider the following question: why is using a robot with several manipulators with complex dynamics and limitations more effective than using a robot with one manipulator of higher power and carrying capacity?

No additional rationale is needed to move objects with large geometries relative to the size of the robot, as the interaction of multiple arms increases motion stability and payload capacity. The problem is to justify the use of two or more manipulators to move a concentrated mass (a load of insignificant size relative to the robot and manipulators). Some studies have shown that the use of robots with coordinated (cooperative) control increases the carrying capacity even for concentrated mass loads [15,16,17,18]. Therefore, the present study considers the problem of a coordinated operation of several underwater robotic arms when transporting an object with a concentrated mass in a limited working space. The control scheme implies different ways to distribute the load between the manipulators.

Because the underwater manipulator is a complex system, highly non-linear, and vulnerable to perturbations caused by underwater influences, reliable control strategies must be developed for the manipulator to obtain a reliable system.

There are several methods to synthesize control for a complex of robots; for example, for the sliding mode, [19,20] suggests using a control and position controller, and [21] proposes an additional verification of the results by Lyapunov functions. The PID control law is considered in [22]. However, these methods are closer to the classical techniques and are not always effective in these circumstances because the quality of manipulator control is easily affected in the underwater working environment. In [23], an intelligent controller based on a fuzzy neural network RBF (radial basis function) method is proposed for accurate and stable control of an underwater robot. Similar methods using neural networks and fuzzy logic are presented in [24,25,26]. For underwater robots, the technique of control synthesis using a nonlinear behavior prediction model is also acceptable [27].

As the basic method of synthesizing controls in this study, we propose to use the representation of nonlinear models of the manipulators’ dynamics in the linear form of the state-space with state-dependent parameters (state-dependent coefficients techniques). That makes it possible to build simple (close to linear) trajectory controllers for manipulator grasping modules in the form of feedback using the so-called state-dependent differential Riccati equation (SDDRE) method.

Finally, it should be noted that the proposed control scheme can be implemented as a centralized control algorithm for the whole system, as well as in a decentralized form, because the algorithm involves the possibility of calculating control signals for each manipulator separately without exchanging data with each other based on information obtained exclusively from their sensors (for example, position and speed measurements). This significantly increases the reliability of the coordinated control system.

The rest of the paper is organized as follows. Section 2 discusses the features of the proposed methods to solve the problem statement, while Section 3 presents applications of the above methods, as well as a test case. In Section 4, the results and analysis are presented. The discussion is presented in Section 5. Finally, the conclusions are given in Section 6.

## 2. Materials and Methods

In the present paper, a cooperative control approach based on the solution of the state-dependent differential Riccati equation is proposed. It should be noted that it is not the only way to obtain control for a cooperative system, but it is optimal, which highlights its advantages among fuzzy logic methods or classical techniques [1,15,28].

Furthermore, the paper applies the well-known algorithm for solving the inverse kinematics problem of robot manipulators, which is based on the pseudo-inversion of the geometric Jacobian [29].

One of the main aspects of solving the problem of mathematical modeling and, in particular, the verification of control laws is the synthesis of the system dynamics model.

The majority of the known scientific research in the field of underwater vehicle modeling is based on the publications of T. Fossen [30]. They include models in 3, 4, and 6 degrees of freedom (DoF), description of external disturbances, control systems, and general calculation procedures. The mathematical model is a fundamental element in path planning, control, and navigation; therefore, this problem is extremely important and complicated in solving the problems identified in the article.

Besides identifying dynamic parameters in control synthesis, it is necessary to solve the problem of inverse dynamics, which allows obtaining the desired forces from the side of manipulating robots. The paper considers the approach of equal load distribution between manipulators, which is described in detail in [31,32].

A substantial contribution to the paper was made by [33], which describes in detail the dynamics and kinematics of the UVMS, the refinements of their design, as well as the specifics of the hydrodynamic parameters’ calculation technique. It is necessary to note that the well-known parametrization approach, presented in [15,34] and which is applicable to the task set in the paper, was used for the composition of the dynamic parameters of the system.

Collectively, all the methods described above form the base for the development of a tracking control system for the cooperative complex of underwater manipulation robots during the transfer of an object with a concentrated mass, which is one of the essential goals of this work.

## 3. Theoretical Part

### 3.1. General Problem Statement

The proposed approach is based on the development of a nonlinear tracking optimal control scheme for the underwater robot–manipulator system UVMS and the object. We assume that all UVMS are informed about both the desired configuration of the object and the position of obstacles in the workspace. Therefore, the control task is to route the entire formation to the target configuration avoiding collisions with static obstacles in the workspace.

Let us consider N underwater robots with manipulators that have m degrees of freedom and rigidly capture an object of mass $m_{O}$ in a limited workspace. Figure 1 shows an example of a cooperative system of two robots with manipulators. Assume that each manipulator is fully driven concerning the coordinate system associated with the working tool (gripper) and the manipulators are equipped with appropriate sensors to measure the position and speed of the manipulator links. In addition, the geometric parameters of both robots with manipulators and the object of manipulation are known.

Motion, trajectory, and coordinates of the manipulator joints are measured relative to the main reference frame $\left\{X_{0},Y_{0},Z_{0}\right\}$. The center of mass (CoM) coordinates of the manipulated object is denoted as $x_{O},y_{O},z_{O}$, the center coordinates of the manipulator working tools (grippers)—as $x_{i},y_{i},z_{i}$ when i=1,2,…,N. The load is considered as a concentrated mass; therefore, the trajectories of the load and the working bodies are similar, i.e., $\left(x_{O},y_{O},z_{O}\right)=\left(x_{i},y_{i},z_{i}\right)$. In the case of a load with a concentrated mass, therefore, the knowledge of the gripper orientation is not required. As a result, the minimum degree of freedom of the manipulators for designing the trajectory of planar motion is two, and for spatial motion it is three.

The task is to design distributed control protocols for each of the N manipulators to safely transport the object along the desired trajectory, considering the desired configuration for the manipulated object movement.

### 3.2. Mathematical Model of the System

#### 3.2.1. The Dynamics Model of the Manipulated Object

Without loss of generality, consider the following model of dynamics for an underwater object, which can be derived on the basis of the Newton–Euler formulations [31,33]:(1)$$or{\dot{p}}_{O}=J_{O}{\left(\eta_{2,O}\right)}^{-1}\upsilon_{O},\;M_{O}\left(p_{O}\right){\dot{\upsilon }}_{O}+C_{O}\left(\upsilon_{O},p_{O}\right)\upsilon_{O}+D_{O}\left(\upsilon_{O},p_{O}\right)\upsilon_{O}+g_{O}=-f_{e},$$
where $p_{O}=\left[\eta_{1,O}^{T},\eta_{2,O}^{T}\right]$, $\eta_{1,O}^{T}={\left[x_{O},y_{O},z_{O}\right]}^{T}$ and $\eta_{2,O}^{T}={\left[\varphi_{O},\theta_{O},\psi_{O}\right]}^{T}$, $M_{O}$—is a positively defined inertia matrix of the object, $C_{O}$—is the Coriolis matrix, and $D_{O}$ models dissipative effects. The gravity and buoyancy vector is denoted as $g_{O}$. The matrix $J_{O}\left(\eta_{2,O}\right)$—is the Jacobian of the object, which converts the derivative of the Euler angles to the angular velocity $\omega_{O}$ (angular velocity in the system coupled to the object); it can be defined as:(2)$$J_{O}\left(\eta_{2,O}\right)=\left(\begin{array}{cc} I_{3\times 3} & 0_{3\times 3}\\ 0_{3\times 3} & J_{O}^{I}\left(\eta_{2,O}\right) \end{array}\right),\;J_{O}^{I}\left(\eta_{2,O}\right)=\left(\begin{array}{ccc} 1 & 0 & -sin(\varphi_{O})\\ 0 & cos(\varphi_{O}) & cos(\theta_{O})sin(\varphi_{O})\\ 0 & -sin(\varphi_{O}) & cos(\theta_{O})cos(\varphi_{O}) \end{array}\right).$$

The external force fe∈ℝ6, acting on the object must be distributed among the manipulators fe=∑i=1Nfe,i, in order to isolate the effect fe,i from each manipulator.

#### 3.2.2. Kinematic Model of a Robot with Manipulators

We define the coordinates of the working organ of each manipulator as pi=[η1,piT,η2,piT], where η1,piT=[xpi,ypi,zpi]T and η2,piT=[φpi,θpi,ψpi]T denote the position and orientation expressed in Euler angles relative to the inertial reference system {X0,Y0,Z0}. Let us also determine the generalized velocities of the manipulator’s working tool through υi=[η˙1,iT,ωiT]T, where η˙1,iT—is the linear velocity, ωi—is the angular velocity. Suppose qi∈ℝm describes the position and orientation of the carrier robot, qi∈ℝm is a vector of generalized coordinates of manipulator joints positions.

The position and orientation of the manipulator grip are given by the direct kinematics as follows:(3)$$p_{i}=F_{i}\left(q_{i}\right),\;\;\;i=1,2\ldots ,N.$$

For the differential kinematics of each manipulator, we have
(4)$$\upsilon_{i}=J_{i}\left(q_{i}\right){\dot{q}}_{i},i=1,2\ldots ,N,\;{\dot{q}}_{i}={\left[{\dot{q}}_{Vi}^{T},{\dot{q}}_{Mi}^{T}\right]}^{T},$$
where ${\dot{q}}_{i}$ is the velocity vector of the manipulator joint coordinates and $J_{i}\left(q_{i}\right)$—is the geometric Jacobian [28,31]. Note that the matrix $J_{i}\left(q_{i}\right)$ becomes singular at the points of kinematic specifics (singularities) defined by the set
(5)$$S_{i}=\left\{q_{i}\in R^{n_{i}}:det(J_{i}\left(q_{i}\right)J_{i}\left(q_{i}{)}^{T}\right)=0\right\},\;\;i=1,2\ldots ,N.$$

The task of identifying the geometric Jacobian can be varied, but the well-known approach is based on vector multiplication and the application of transformation matrices [35].

The number of rows in the Jacobian matrix equals the number of states in the velocity vector, while the number of columns is dependent on the number of joints in the UVMS:(6)$$J_{i}\left(q_{i}\right)=\left[\begin{array}{ccc} J_{1}\left(q_{i}\right) & \cdots & J_{m+6}\left(q_{i}\right) \end{array}\right].$$

The components of angular and linear velocities are defined by the following expressions
(7)$$J_{i}\left(q\right)=\left[\begin{array}{c} J_{v_{i}}\left(q\right)\\ J_{\omega_{i}}\left(q\right) \end{array}\right]=\left[\begin{array}{c} {𝓏}_{i-1}\times \left(o_{m+6}-o_{i-1}\right)\\ {𝓏}_{i-1} \end{array}\right]—\mathrm{if}\;i\;\mathrm{link}\;\mathrm{is}\;\mathrm{rotational},\;J_{i}\left(q\right)=\left[\begin{array}{c} J_{v_{i}}\left(q\right)\\ J_{\omega_{i}}\left(q\right) \end{array}\right]=\left[\begin{array}{c} {𝓏}_{i-1}\\ 0_{3\times n} \end{array}\right]—\mathrm{if}\;i\;\mathrm{link}\;\mathrm{is}\;\mathrm{prismatic},$$
where $o_{i-1}$—the first three elements of the fourth column of the matrix $T_{i-1}\left(q\right)$, ${𝓏}_{i-1}$—the first three elements of the third column of the matrix $T_{i-1}\left(q\right)$.

#### 3.2.3. Dynamics Model of an Underwater Robot with Manipulators

The dynamics of underwater robot manipulators can be written as [31,33]:(8)$$M\left(q\right)\ddot{q}+C\left(\dot{q},q\right)\dot{q}+D\left(\dot{q},q\right)\dot{q}+g\left(q\right)=u+J^{T}\left(q\right){\bar{f}}_{e},$$
where $q={\left[q_{1}^{T},\ldots ,q_{N}^{T}\right]}^{T}\in R^{Nm}$, $u={\left[u_{1}^{T},\ldots ,u_{N}^{T}\right]}^{T}\in R^{Nm}$, $M=diag\left\{M_{i}\right\}\in R^{Nm\times Nm}$, $C=diag\left\{C_{i}\right\}\in R^{Nm\times Nm}$, $D=diag\left\{D_{i}\right\}\in R^{Nm\times Nm}$, $g={\left[g_{1}^{T},\ldots ,g_{N}^{T}\right]}^{T}\in R^{Nm}$, ${\bar{f}}_{e}=\left[f_{e,1}^{T},f_{e,2}^{T},\ldots ,f_{e,N}^{T}\right]$.

The trajectory of the manipulators is predetermined by the desired motion of the object; therefore, Equation (1) is a known equation, but the values of $f_{e,i}$, i=1,2,…,N are undefined. To solve this problem and divide $f_{e,i}$ between the robots, one can apply the optimal load distribution method [32].

Assume that the robotic arms are rigidly gripping the object. Due to the rigid grip of the object, the following equations are satisfied:(9)pi=pO+[RIOliαi],i=1,2,…,N,
where vectors $l_{i}={\left[l_{ix},l_{iy},l_{iz}\right]}^{T}$ and $\alpha_{i}={\left[\alpha_{ix},\alpha_{iy},\alpha_{iz}\right]}^{T}$ represent the constant relative position and orientation of the manipulator working tool relative to the object, expressed in the object frame and RIO denotes the rotation matrix, which describes the orientation of the object, expressed in the inertial reference frame {I}. Therefore, using (9) for each manipulator, it is possible to calculate the position of the object relative to the inertial reference frame {I}, because the geometric parameters of the object are considered to be known.

In addition, the kinetostatic duality along with the gripping rigidity assumes that the force $f_{e}$, acting on the object’s center of mass and the generalized forces $f_{e,i}$, acting on the manipulators at the gripping points are related through [32].
(10)$$f_{e}=\left[\begin{array}{cccc} J_{O1}^{T} & J_{O2}^{T} & \ldots & J_{ON}^{T} \end{array}\right]{\left[\begin{array}{cccc} f_{e,1}^{T} & f_{e,2}^{T} & \ldots & f_{e,N}^{T} \end{array}\right]}^{T}=\;=J_{O1}^{T}f_{e,1}+J_{O2}^{T}f_{e,2}+\ldots +J_{ON}^{T}f_{e,N}=\sum_{i=1}^{N}J_{Oi}^{T}f_{e,i},J_{Oi}=J_{iO}^{-1}.$$

Now, consider the constants $k_{i}$, i=1,2,…,N satisfying the $k_{1}+k_{2}+\ldots +k_{N}=1$ condition, which we enter as load distribution coefficients between manipulators.

Given $k_{1}+k_{2}+\ldots +k_{N}=1$, the dynamics of the object (1) can be rewritten as (the dependence in the dynamics matrices on the number is omitted to simplify the notation):(11)$$\sum_{i=1}^{N}k_{i}\left\{M_{O}{\dot{\upsilon }}_{O}+C_{O}\upsilon_{O}+D_{O}\upsilon_{O}+g_{O}\right\}=-\sum_{i=1}^{N}J_{Oi}^{T}f_{e,i}.$$

From the last relation, we can calculate the forces for each of the manipulators
(12)$$f_{e,i}=-k_{i}J_{iO}^{T}f_{e}.$$

Once you have the load distribution scheme between the manipulators, you can develop a controller to control the entire system (8).

### 3.3. Control Algorithm

Figure 2 demonstrates a scheme of the UVMS tracking control, which clearly shows the separation of the applied methods described in the second section. Each block has its own task, and together they form a complete architecture of a cooperative control system. The used methodology, encapsulated in each block, is described in the second section of the paper. Therefore:

«Desired object trajectory» generates the desired trajectory of the object’s center of mass $p_{O}$;«Inverse Dynamics» calculates the desired forces $f_{e,i}$, applied to the manipulation robots;«SDDRE Tracking Control» produces optimal control itself;«$UVMS_{i}$» is a dynamics model of the underwater vehicle–manipulator system;«Desired end-effector position» calculates the desired position of the UVMS tool according to a given trajectory and relative end-effector position;«Kinematic Control» transposes from the end-effector’s Cartesian coordinate system $p_{i,des}$ into the space of the manipulator’s generalized coordinates $q_{i,des}$.

The control synthesis algorithm can be written as a set of the following steps:

Setting the desired trajectory of the object $p_{O}=\left[\eta_{1,O}^{T},\eta_{2,O}^{T}\right]$;The inverse dynamics problem is solved on the basis of the motion model of the object (1). Knowing the motion, determine which force $f_{e}=\overset{N}{\underset{i=1}{\sum }}f_{e,i}$ should be applied to the object to provide the desired movement. After that, this force is distributed between manipulators in proportion according to (11) and (12);Based on the desired motion of the object, the problem of determining the desired motion of the working tools of manipulators is solved through (9). It is necessary to set the parameters of the relative position of the object and the manipulator grips, i.e., to set vectors for each manipulator $l_{i}={\left[l_{ix},l_{iy},l_{iz}\right]}^{T}$ and $\alpha_{i}={\left[\alpha_{ix},\alpha_{iy},\alpha_{iz}\right]}^{T}$;Knowing the required motions of the manipulator working tools, the problem of kinematic control is solved using the given motion of the working tool of the manipulator to determine the required value of the generalized coordinates of the robot–manipulator system, i.e., a kinematic regulator is constructed (Figure 3);Knowing the desired value of the generalized coordinates and the required force for each manipulator, the problem of synthesizing a tracking SDDRE controller is solved.

There are a number of methods to find the inverse Jacobian for solving the inverse kinematics problem; however, in this paper, we will consider the so-called damped least-squares (DLS) inverse method proposed in [29]:(13)$$J^{-1}\left(q_{i}\right)=J^{T}\left(q_{i}\right){\left[J\left(q_{i}\right)\cdot J^{T}\left(q_{i}\right)+k^{2}I\right]}^{-1},$$
where k is a damping factor that renders the inversion better conditioned from a numerical viewpoint.

### 3.4. Synthesis of an SDDRE Tracking Controller

To synthesize the controller while solving the problem of tracking the desired value of the manipulator’s generalized coordinates, we use the SDDRE method, which has proven itself in many applications, including the control of manipulators [36].

We denote $x={\left[{\begin{array}{cc} q^{T} & \dot{q} \end{array}}^{T}\right]}^{T}$ and rewrite the nonlinear system (8) with new notations:(14)$$\dot{x}=\left(\begin{array}{c} \dot{q}\\ M{\left(q\right)}^{-1}\left(u+J^{T}\left(q\right){\bar{f}}_{e}-C\left(\dot{q},q\right)\dot{q}-D\left(\dot{q},q\right)\dot{q}-g\left(q\right)\right) \end{array}\right).$$

The latest equation can be represented in the form of state-space equations with state-dependent coefficients (SDC form)
(15)$$\frac{dx}{dt}=A\left(x,t\right)x+B\left(x,t\right)\bar{u},x\left(t_{0}\right)=x_{o},\;A\left(x,t\right)x=\left(\begin{array}{cc} 0_{Nm\times Nm} & I_{Nm\times Nm}\\ 0_{Nm\times Nm} & -M{\left(x\right)}^{-1}\left(C\left(x\right)+D\left(x\right)\right) \end{array}\right),\;B\left(x,t\right)=\left(\begin{array}{c} 0_{Nm\times Nm}\\ M{\left(x\right)}^{-1} \end{array}\right),\;\bar{u}=u+u_{add}\left(x\right),\;u_{add}=J^{T}\left(x\right){\bar{f}}_{e}-g\left(x\right).$$

Note that representation (15) is not unique, and there are different possibilities for the representation of the system matrices. The problem of finding the optimal representation for the system matrices is called the parameterization problem (or factorization), and there are different approaches to its solution. For example, it is possible to select the structure and parameters of the matrices based on preserving the controllability requirements of the system. The parametrization problem itself is rather complicated, so these aspects will not be addressed here.

Consequently, for the obtained system (15), we formulate the problem of the criterion minimization (here and further, we will omit the dependence on time where it is not essential for shortening the notation)
(16)$$S_{t_{0}t_{f}}\left(u\right)=\frac{1}{2}e^{T}\left(t_{f}\right)Fe\left(t_{f}\right)+\frac{1}{2}\int_{t_{0}}^{t_{f}}\left[e^{T}Q\left(x\right)e+u^{T}R\left(x\right)u\right]dt,\;Q\left(x\right)\geq 0,R\left(x\right)>0,F>0.$$
where $e=x-x_{ref}$, $x_{ref}$—is the desired value of the system state vector (14), which is determined by the desired value of the generalized coordinates.

If the pair of matrices (A(x,t), B(x,t)) is pointwise controllable for all x(t) for $t\in \left[t_{0},t_{f}\right]$, the solution of problem (16) in the form of feedback can be obtained using the SDDRE method.

### 3.5. Centralized Control Scheme

The solution of problem (16) for feedback control can be written in the same way as for the linear case [34]:(17)$$u=-R^{-1}\left(x\right)B^{T}\left(x,t\right)\left(K\left(x\right)x+g\left(x,t\right)\right),$$
where $\dot{K}\left(x\right)=-A{\left(x\right)}^{T}K\left(x\right)-K\left(x\right)A\left(x\right)-Q\left(x\right)+K\left(x\right)B\left(x\right)R{\left(x\right)}^{-1}B^{T}\left(x\right)K\left(x\right)$, $K\left(t_{f}\right)=F$, $\dot{g}=-{\left(A\left(x\right)-B\left(x\right)R{\left(x\right)}^{-1}B^{T}\left(x\right)K\left(x\right)\right)}^{T}g+Q\left(x\right)r,g\left(t_{f}\right)=Fx_{ref}\left(t_{f}\right)$.

Two approaches to the implementation of this solution are possible: solving N-times SDDRE type control (17) for each manipulator independently and solving SDDRE control once for the whole system. These approaches were proposed in [17].

This numerical solution requires integration of Equation (17) in the opposite direction from time $t_{f}$ to t, which is not realizable because the state x of the system (15) at future time moments is not defined and, therefore, the state-dependent parameters of matrices A(x,t), B(x,t) are also unknown.

A Lyapunov-based approach solves the problem in one forward round and was recommended for a tracking problem [34]. Let time tend to infinity, then K in infinity for (17) results as a constant gain, and the steady-state form of K is reformed to the state-dependent Riccati equation (SDRE). As a result, we have a Lyapunov equation with state and control nonlinearities with the final boundary condition at the right time. The solution of this equation results in the suboptimal gain as
(18)$$K\left(x\right)=K_{ss}\left(x\right)+P^{-1}\left(x\right),$$
where $P\left(x\right)=E\left(x\right)+e^{A_{cl}\left(x\right)\left(t-t_{f}\right)}\left(P\left(t_{f}\right)-E\left(x\right)\right)e^{A_{cl}^{T}\left(x\right)\left(t-t_{f}\right)}$, $A_{cl}$ is a closed-loop system matrix, $K_{ss}$ is a negative root of SDRE.

The closed-form solution for g for (17) is presented as follows. This equation is rewritten as
(19)$$\dot{g}=-A\left(x\right)g+Q\left(x\right)r.$$
and the solution has the form of
(20)$$g=A^{-1}\left(x\right)\left(Q\left(x\right)r+e^{A\left(x\right)\left(t_{f}-t\right)}A\left(x\right)g\left(t_{f}\right)-Q\left(x\right)r\right).$$

The proposed closed-form answer works perfectly. Nevertheless, when the end of the trajectory is not zero, oscillation and an unacceptable jump occur. The cause of this problem is an open question.

### 3.6. Decentralized Control Scheme

In the first case, we need to calculate the control law for each UVMS
(21)$${\bar{u}}_{i}=-R_{i}^{-1}\left(x_{i}\right)B_{i}^{T}\left(x_{i},t\right)K_{i}\left(x_{i},t\right)x_{i}+u_{i,add},$$
where $x_{i}\left(t\right)$—are the states of the i manipulator. Matrix $K_{i}\left(x_{i},t\right)$ is obtained from the above algorithm, where instead of block matrices A(x), B(x), R(x), and Q(x), the corresponding matrices for each manipulator are used separately, and the model of manipulator dynamics in SDC form is
(22)$$\frac{dx_{i}}{dt}=A_{i}\left(x_{i},t\right)x+B_{i}\left(x_{i},t\right){\bar{u}}_{i},\;x_{i}\left(t_{0}\right)=x_{i,o},\;A_{i}\left(x_{i},t\right)x_{i}=\left(\begin{array}{cc} 0_{m\times m} & I_{m\times m}\\ 0_{m\times m} & -M_{i}^{-1}\left(x\right)\left(C_{i}\left(x\right)+D_{i}\left(x\right)\right) \end{array}\right),\;B_{i}\left(x_{i},t\right)=\left(\begin{array}{c} 0_{m\times m}\\ M_{i}^{-1}\left(x_{i}\right) \end{array}\right),\;{\bar{u}}_{i}=u_{i}+u_{i,add}\left(x_{i}\right),\;u_{i,add}=J_{i}^{T}\left(x_{i}\right)f_{i,e}-g_{i}\left(x_{i}\right),\;i=1,2,\ldots ,N.$$

The matrix K(x,t) of the controller (17) when solving the control problems separately for each vehicle manipulator by analogy with [37] can be composed of elements $K_{i}\left(x_{i},t\right)$ in the form
(23)$$K\left(x,t\right)=\left(\begin{array}{ccccc} K_{1}\left(x_{1},t\right) & 0 & 0 & 0 & 0\\ 0 & K_{2}\left(x_{2},t\right) & 0 & \vdots & 0\\ 0 & 0 & \ddots & 0 & \vdots \\ \vdots & \vdots & 0 & K_{N-1}\left(x_{N-1},t\right) & 0\\ 0 & 0 & \cdots & 0 & K_{N}\left(x_{N},t\right) \end{array}\right).$$

In this case, the system has N independent subcontrollers for each vehicle manipulator separately.

The second approach uses parameterized matrices for the entire system (15), requires solving SDDRE only once, and implements the control law (17). This approach provides optimal control of the complete system.

### 3.7. Test Case

#### 3.7.1. Object Dynamics

In order to verify the method, consider a specific test case. As an object of manipulation, a body in the form of a cylindrical rod (Figure 4) of mass $m_{O}=23$ kg, length $l_{O}=2.5$ m, and radius $r_{O}=0.05$ m is used.

The tensor of inertia of the handled object has the following definition
(24)$$I_{O}=\left(\begin{array}{ccc} \frac{1}{12}m_{O}l_{O}^{2} & 0 & 0\\ 0 & 0 & 0\\ 0 & 0 & \frac{1}{12}m_{O}l_{O}^{2} \end{array}\right).$$

Since the working environment is underwater, it is necessary to take into account the hydrodynamic effects. The added masses of the rod, according to [5], are defined as
(25)$$X_{\dot{u}}=-0.1m_{O},\;Y_{\dot{v}}=-\pi \rho r_{O}^{2}l_{O},\;Z_{\dot{w}}=-\pi \rho r_{O}^{2}l_{O},\;K_{\dot{p}}=0,\;M_{\dot{q}}=-\frac{1}{12}\pi \rho r_{O}^{2}l_{O}^{3},\;N_{\dot{r}}=-\frac{1}{12}\pi \rho r_{O}^{2}l_{O}^{3},$$
where ρ=1025 and represents the density of the fluid environment (salted water).

#### 3.7.2. UVMSs Dynamics

As an example of the method’s operability, consider the situation depicted in Figure 1. The work [38] presents the dynamic parameters of the Sf-30k underwater vehicle (Figure 5), which we use to simulate the mobile manipulator bases.

Dynamics matrices for two identical Sf-30k with consideration of hydrodynamic effects are given below.
(26)$$M_{1}=\left[\begin{array}{cccccc} 2798 & 0 & 0 & 0 & 872.9 & 0\\ 0 & 3314 & 0 & -872.9 & 0 & 0\\ 0 & 0 & 4831 & 0 & 0 & 0\\ 0 & -872.9 & 0 & 942.1 & 0 & 0\\ 872.9 & 0 & 0 & 0 & 1997 & 0\\ 0 & 0 & 0 & 0 & 0 & 1889 \end{array}\right];$$
(27)$$D_{1}\left(v\right)=D_{LIN}+D_{QUAD}\left(v\right);\;D_{QUAD}\left(v\right)=\left[\begin{array}{cccccc} 773.6\left|u\right| & 0 & 0 & 0 & 0 & 0\\ 0 & 1067\left|v\right| & 0 & 0 & 0 & 0\\ 0 & 0 & 2277\left|w\right| & 0 & 0 & 0\\ 0 & 0 & 0 & 302.5\left|p\right| & 0 & 0\\ 0 & 0 & 0 & 0 & 1078\left|q\right| & 0\\ 0 & 0 & 0 & 0 & 0 & 558.1\left|r\right| \end{array}\right];\;D_{LIN}=\left[\begin{array}{cccccc} 123.8 & 0 & 0 & 0 & 0 & 0\\ 0 & 170.7 & 0 & 0 & 0 & 0\\ 0 & 0 & 364.3 & 0 & 0 & 0\\ 0 & 0 & 0 & 129.1 & 0 & 0\\ 0 & 0 & 0 & 0 & 188.0 & 0\\ 0 & 0 & 0 & 0 & 0 & 97.36 \end{array}\right];$$
(28)$$C_{1}\left(v\right)=\left[\begin{array}{cccccc} 0 & 0 & 0 & 0 & 4831w & -3314v\\ 0 & 0 & 0 & -4831w & 0 & 2798u\\ 0 & 0 & 0 & 3314v & -2798u & 0\\ 0 & 4831w & -3314v & 0 & 0 & 0\\ -4831w & 0 & 2798u & -18 & 0 & 0\\ 3314v & -2798u & 0 & 0 & 0 & 0 \end{array}\right];$$
(29)$$g_{1}\left(\eta \right)=\left[\begin{array}{c} 0\\ 0\\ 0\\ 7081cos\theta sin\phi \\ 7081sin\theta \\ 0 \end{array}\right].$$

Matrices (26)–(29) already contain hydrodynamic effects calculated according to the strip theory [38].

The six-link robot (Figure 6) with cylindrical rotating links as manipulative machines [35] will be considered below. Table 1 shows the parameters of both devices.

Workclass ROVs are often equipped with one or two manipulators, depending on the type of operation the ROV is designed for. The main reason for equipping ROVs with manipulators is to design them for intervention tasks. By mounting the ROV with manipulators, additional hydrodynamic, hydrostatic, and rigid body inertia forces and moments will affect the vehicle [5,32]. If the ROV is large enough, these effects are negligible. However, these effects usually have to be modelled and taken into account when designing workclass ROVs. Here, the manipulator is of considerable size, and the presence of the manipulator will create forces and moments on the ROV that affects the response of the system. In addition, when the manipulators perform intervention tasks, they come in contact with the environment, which creates contact forces that also will affect the ROV. No matter the size difference between the ROV and the manipulator, the contact forces can be of considerable size, which needs to be taken into account when modelling the system.

The joint motion is often a very simple one degree of freedom linear or rotational motion. Manipulator systems therefore consist of either revolute or prismatic joints or a combination of these. The manipulator system considered in this paper, shown in Figure 6, is built up by revolute joints. This gives very favorable characteristics to the robot’s workspace, and it makes the modelling a lot easier, as the homogeneous transformation matrices can be written in a very simple form using D-H parameters in Table 2.

The resulting equations are quite cumbersome, but the method of calculating the manipulator’s dynamic parameters based on the Euler–Lagrange method is redundantly described in papers [35,37,38]. The source [33] provides formulas to calculate the hydrodynamic effects for the cylindrical shape, which were used to calculate the resulting parameters.

It is necessary to take into account the offset of the manipulator’s base relative to the center of mass of the vehicle. The resulting required transformation matrix, which provides all the necessary basic kinematic transformations, is shown below
(30)$$T_{0}=\left[\begin{array}{cccc} 1 & 0 & 0 & 0.75\\ 0 & 1 & 0 & 0\\ 0 & 0 & 1 & -0.75\\ 0 & 0 & 0 & 1 \end{array}\right].$$

From this, one can see the desired offset in the last column of the matrix (30).

#### 3.7.3. Controller Parameters

One of the most important tasks in the synthesis of SDDRE control is the selection of penalty matrices Q and R. There are no clearly formulated rules in selecting the numerical values for each of them. This is a creative challenge that requires data calibration, for example, in numerical simulations.

In the context of this paper, the penalty matrices for the first and the second UVMS were matched as follows:(31)$$Q=\left(\begin{array}{cc} 10^{4}\cdot I_{12\times 12} & 0_{12\times 12}\\ 0_{12\times 12} & 0_{12\times 12} \end{array}\right),\;R=10^{-4}\cdot I_{12\times 12}.$$

In this way, parameters (31) restrict the states and provide more freedom to the system control.

## 4. Results

The following section will present the Simulink model applied to simulate the system, and all the relevant blocks will be reviewed. The Simulink model for the underwater vehicle–manipulator system is presented in Figure 7.

The Simulink model consists of kinematic and dynamic modeling blocks for each link. Reference block “Object” define reference values and constants.

Figure 8 shows the subsystem for calculating the generalized coordinates and position of the working tool using Lagrange dynamics and solving the Jacobian velocity conversion task. The outputs of the block are position and vector q.

Figure 9 shows the module for calculating the desired generalized coordinates based on the method proposed in [29] using Equation (13).

The desired trajectory of the controlled object is shown in Figure 10. Here, you can see the desired angular and translational changes in the position of the object’s center of mass. The object should move along a circular trajectory. The result of the first UVMS movement according to the desired trajectory is shown in Figure 11.

Analogous results for the second UVMS are shown in Figure 12. As a result, the trajectory controller works correctly according to the system parameters.

Figure 13 shows that the tracking control of the object occurs with a slight error; however, the endpoint of the trajectory is obtained.

The magnitude of the object’s deviation from the desired trajectory is determined by the RMSE and MAE metrics, which are summarized in Table 3 for each degree of freedom.

## 5. Discussion

The results obtained in the simulation can be ambiguous for the considered method, but the synthesized control is optimal, which is an indisputable advantage in comparison with the classical methods of control systems.

Undoubtedly, the derived calculation methodology is valid, which is proved by the results of the third section of the paper. Practical applications are quite urgent, because there is a tendency for underwater robotics development in the world. In this case, it is necessary to synthesize a control law, which shall satisfy the optimality requirements. The work results can be used as a calculation method for controllers of robotic complexes in the underwater environment.

A further development vector is directed towards the task of comparing centralized and decentralized control systems. Additionally, it is necessary to consider the more complex case of a control system with many robots in a constrained environment with obstacles.

## 6. Conclusions

In this research, an algorithm for coordinated control of a robotic complex with N manipulators while performing the work of moving a concentrated mass cargo along a given trajectory was developed. To synthesize the control, we used an approach based on the representation of nonlinear manipulator dynamics models in the linear state-space form with state-dependent parameters (SDC-form). That allowed us to apply suboptimal control methods with the feedback based on the solution of the SDDRE. It should be noted that the SDDRE method provides the construction of control on a finite time interval and provides the change in control intensity taking into account the transient effect of the system matrices in SDC form.

The research highlights two approaches of system implementation: a common controller for the complete system and a set of N independent subcontrollers for RTC manipulators.

The results of both approaches are similar, but for the systems with little interaction between manipulators, a common structure is recommended, and for the systems with a large number of manipulators, the approach with independent subcontrollers may be more acceptable.

In practical implementation, the failure of one subcontroller does not ruin the performance of the entire system. Therefore, it is recommended to use an independent design for experimental operations in order to maintain greater system safety.

## Figures and Tables

**Figure 1 sensors-22-05038-f001:**
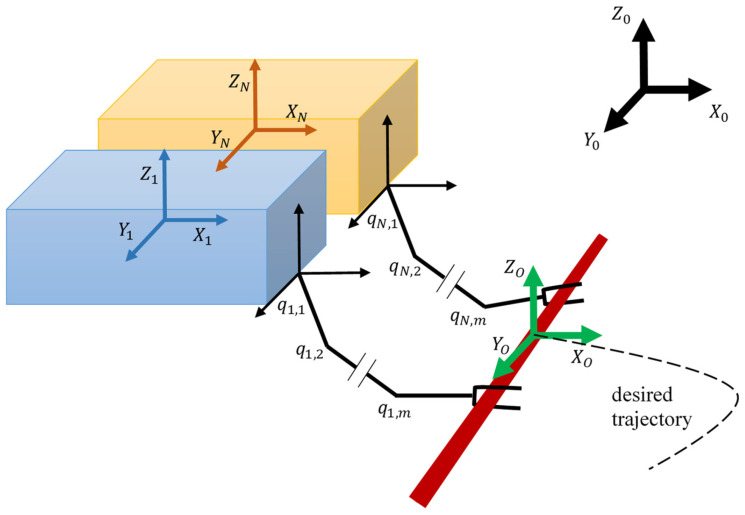
Schematic representation of two underwater robots with manipulators as a cooperative system.

**Figure 2 sensors-22-05038-f002:**
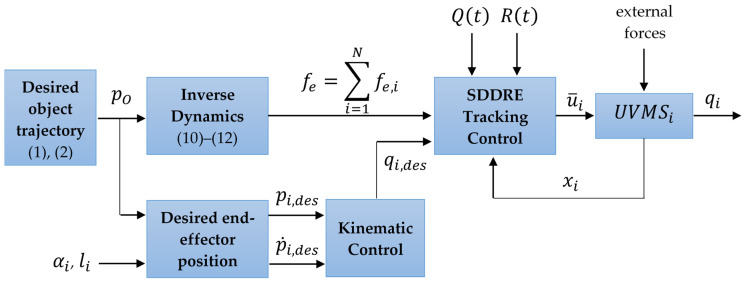
UVMS’s tracking control diagram.

**Figure 3 sensors-22-05038-f003:**
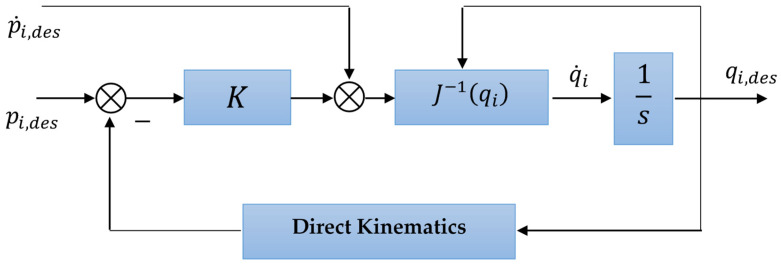
Schematic representation of the kinematic control.

**Figure 4 sensors-22-05038-f004:**
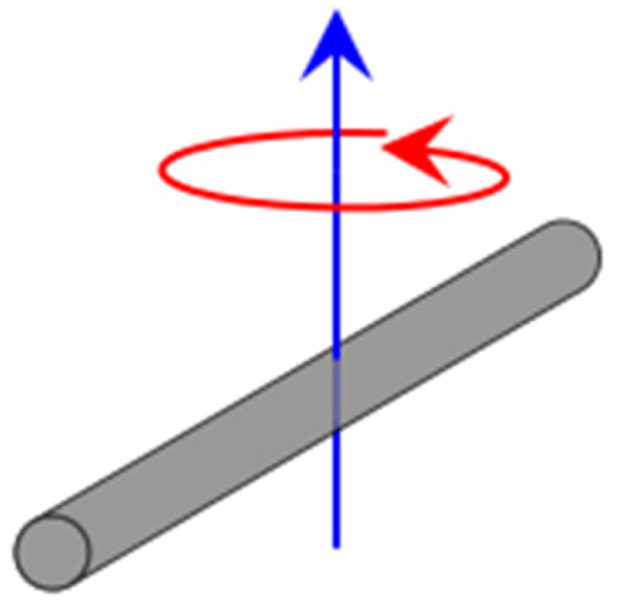
Cylindrical rod object.

**Figure 5 sensors-22-05038-f005:**
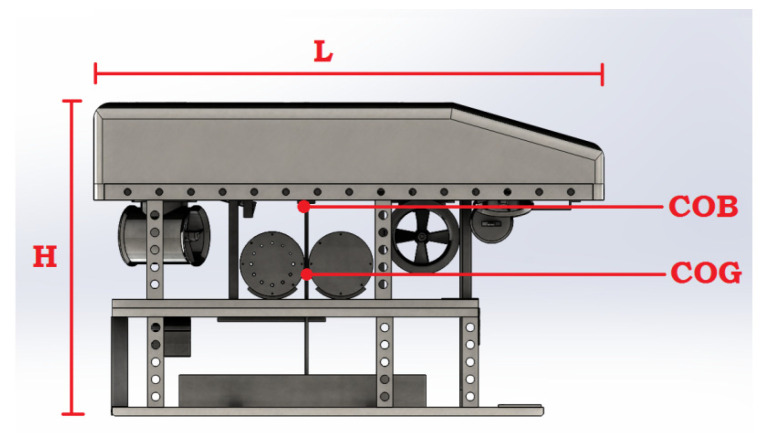
Underwater Robot Sf-30k.

**Figure 6 sensors-22-05038-f006:**
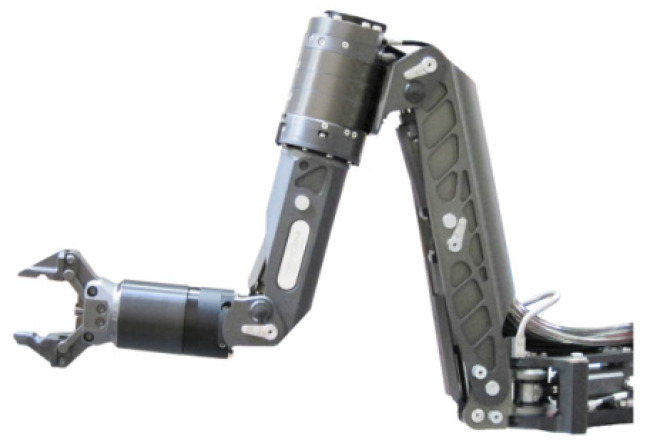
Six-link robot «Orion 7R» kinematic scheme [35].

**Figure 7 sensors-22-05038-f007:**
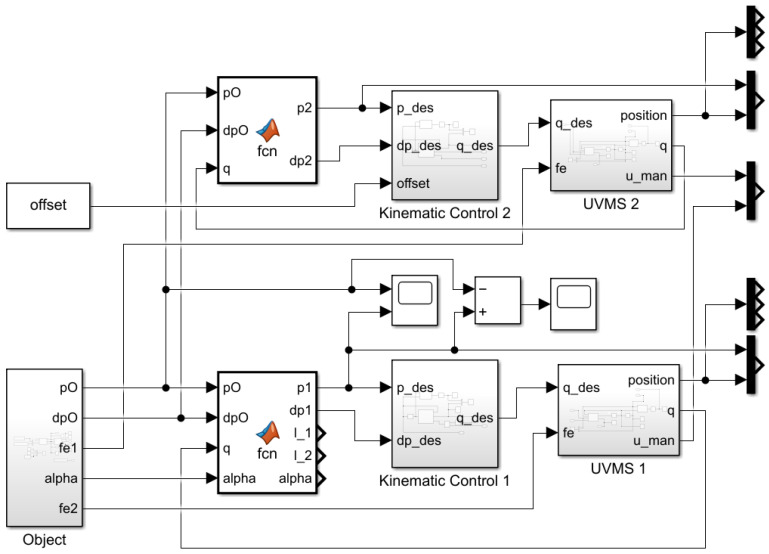
Simulink model for the two UVMSs.

**Figure 8 sensors-22-05038-f008:**
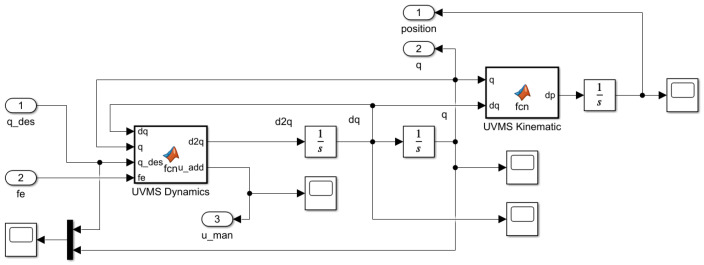
Simulink submodel of UVMS.

**Figure 9 sensors-22-05038-f009:**
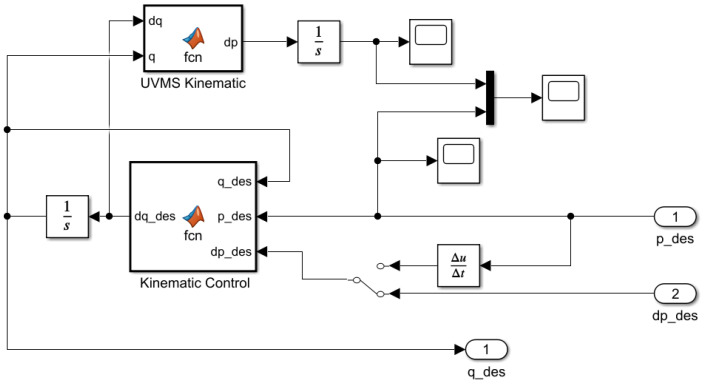
Simulink submodel of kinematic control.

**Figure 10 sensors-22-05038-f010:**
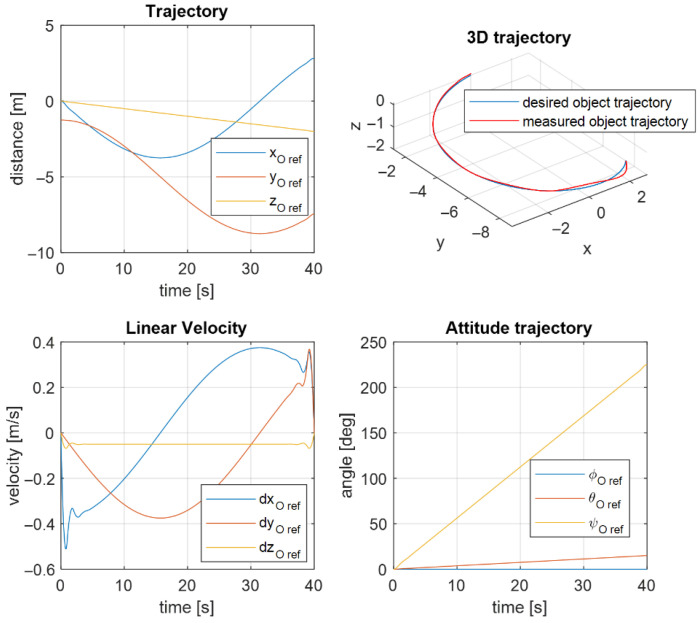
Desired object trajectory.

**Figure 11 sensors-22-05038-f011:**
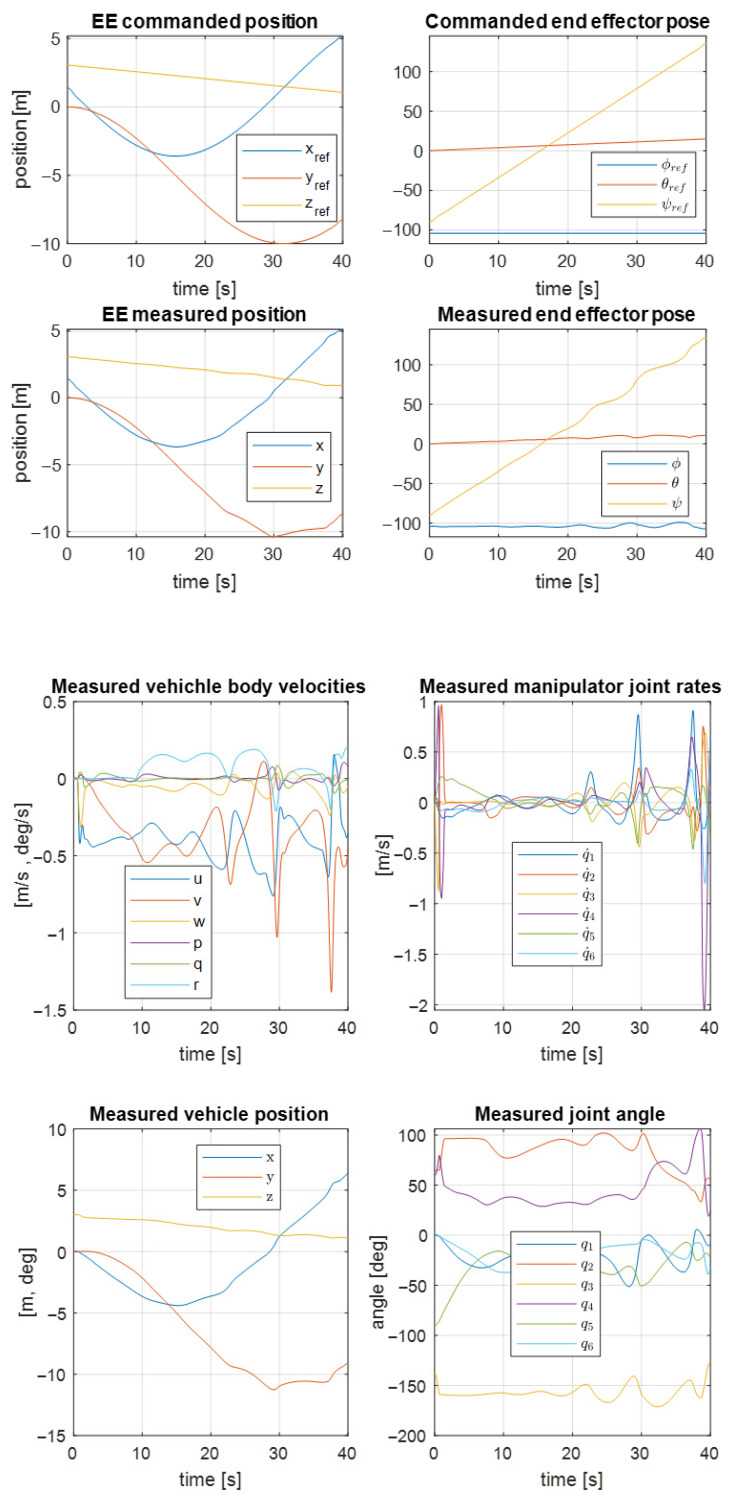
Measured and commanded positions of the first UVMS.

**Figure 12 sensors-22-05038-f012:**
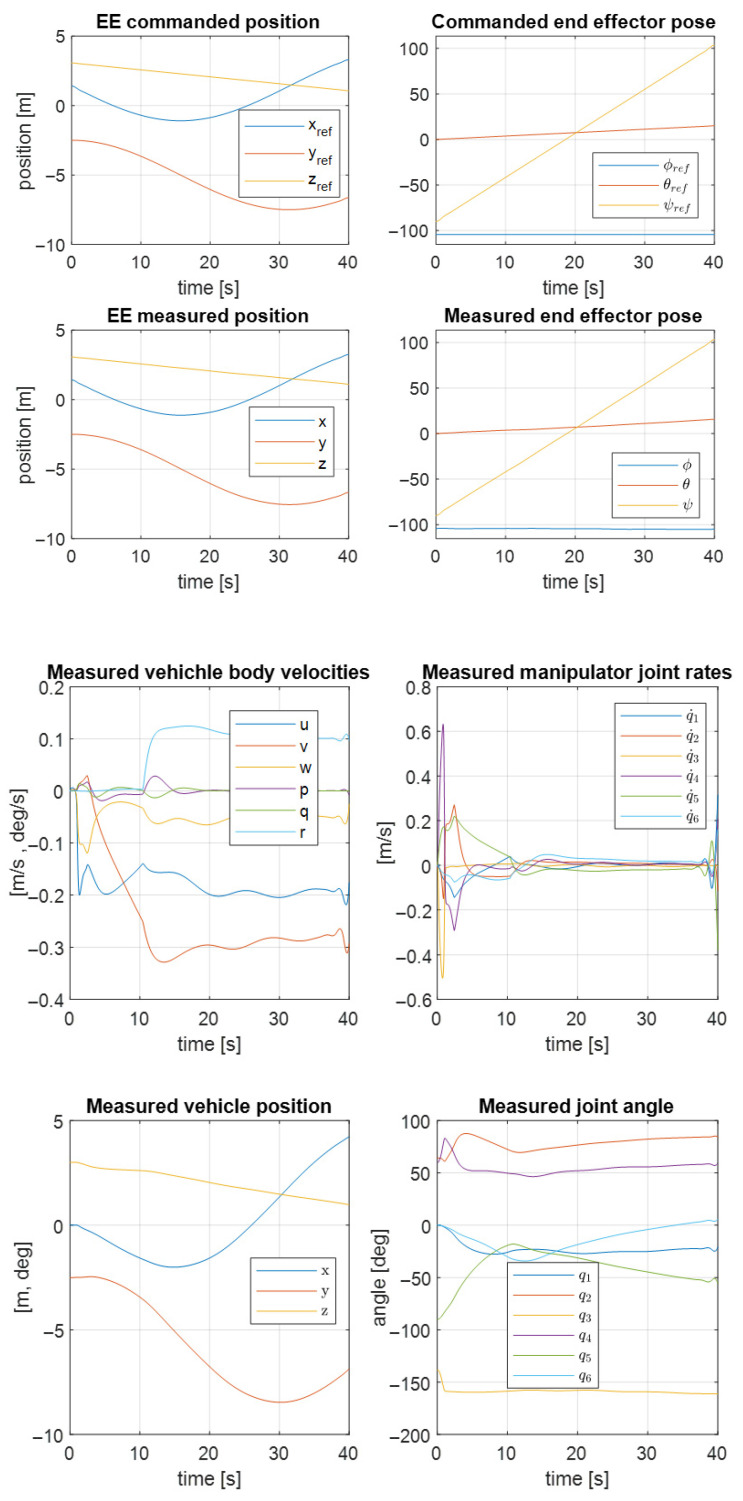
Measured and commanded positions of the second UVMS.

**Figure 13 sensors-22-05038-f013:**
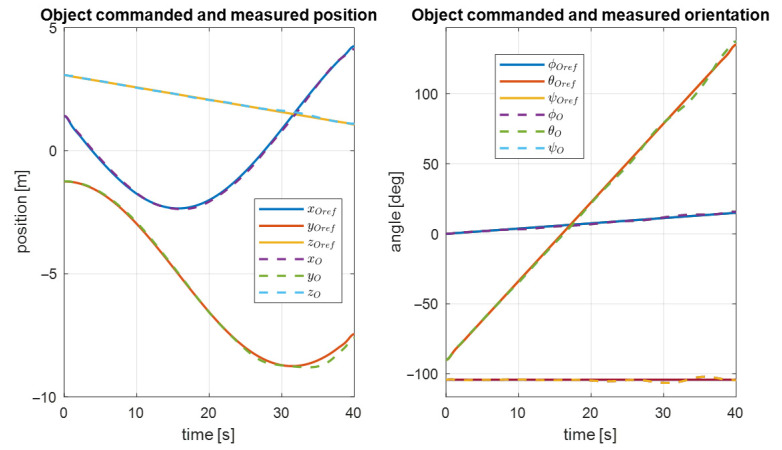
Measured and commanded positions of the object.

**Table 1 sensors-22-05038-t001:** Manipulator link parameters.

Link *i*	Link Mass *m_i_*, (kg)	Buoyancy *B_i_* (N)	Radius *r_i_* (m)	Length *l_i_* (mm)
1	1.50	0.4	0.1	64.9
2	1.25	0.2	0.1	957.1
3	1.25	0.4	0.1	125
4	2.1	0	0.1	580
5	1.6	0	0.1	259
6	1.05	0.15	0.1	0

**Table 2 sensors-22-05038-t002:** D-H parameters.

Link *i*	*a_i_* (mm)	*a_i_* (rad)	*d_i_* (mm)	*q_i_* (rad)
1	64.9	π/2	0	$q_{1}$
2	957.1	0	0	$q_{2}+\pi /2$
3	125	π/2	0	$q_{3}-\pi /2$
4	0	−π/2	580	$q_{4}$
5	259	−π/2	0	$q_{5}$
6	0	−π/2	0	$q_{6}$

**Table 3 sensors-22-05038-t003:** RMSE and MAE metrics of object position/orientation.

DOF	RMSE	MAE
x	0.0221	0.0481
y	0.0357	0.0664
z	0.0106	0.0166
ϕ	0.2515	0.6233
θ	0.0749	0.4101
ψ	0.6386	1.2101

## Data Availability

The data presented in this study are available upon request from the corresponding author.

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
