# Peer review of "Cooperative Control of Underwater Vehicle–Manipulator Systems Based on the SDC Method"

_sensors, 2022, doi:10.3390/s22135038_

Round 1

Reviewer 1 Report

I send the review in the attachment.

Author Response

Response to Reviewer 1 Comments

Point 1: I propose to extend the literature in first paragraph of the introduction, related to the applications of unmanned robotics (UAVs, USVs and UUVs) that are on the Journal Citations Reports (JCR) list.

 Response 1: Thanks for your nice suggestions. The first section of the article expanded on the overview of UVMS as a control object. Moreover, the range of sources concerning unmanned underwater vehicles (UUVs) has been expanded [6-10].

 Point 2: In the introduction, it is worth summarising what the individual chapters of the article contain.

Response 2: Thank you for your precious comments. At the end of the first section there is a brief overview of each section of the paper.

Point 3: Please write sentences impersonally.

Response 3: Thank you for your precious comments.  The work has been revised for personal style. The pronouns "We" have been removed.

Reviewer 2 Report

The authors proposed a cooperative control of underwater vehicle-manipulator systems based on the SDC method. The paper showed some depth of the cooperative control and provided some results. 

This paper lacks in two different aspects:

1) The literature review lacks of depth of the underwater vehicle-manipulator systems and associated techniques. UVMS is a hot research topic in the world but it is not represented here.

2) While the theoretical section was presented in detail (even though I disagree with the title of theoretical part), the results were merely presented as "here it is" without any further interpretations. Errors were not plotted or tabulated. Significance of the similarities (or differences) were thus not highlighted and it is difficult to see the validity of the results.

I highly suggest that the authors strengthen these two areas for this paper to be accepted.

Author Response

Response to Reviewer 2 Comments

Point 1: The literature review lacks of depth of the underwater vehicle-manipulator systems and associated techniques. UVMS is a hot research topic in the world but it is not represented here.

 Response 1: Thank you for your precious comments. The first section of the article expanded on the overview of UVMS as a control object. Moreover, the range of sources concerning unmanned underwater vehicles (UUVs) has been expanded [6-10].

Point 2: While the theoretical section was presented in detail (even though I disagree with the title of theoretical part), the results were merely presented as "here it is" without any further interpretations. Errors were not plotted or tabulated. Significance of the similarities (or differences) were thus not highlighted and it is difficult to see the validity of the results.

Response 2: Thank you for your precious comments. This way of the results presentation is related to the compilation nature of the article. The task was not to produce our own mathematical methods, but to assemble the existing ones into a single coherent algorithm. For this reason, the presented results (including formulas) describe the essence of the topic of the paper with appropriate references to sources.

The article now includes a table of metrics to compare the desired and received trajectories in the form of RMSE and MAE (Table 3).

Reviewer 3 Report

In the abstract, when you present "Here we used the approach based on the representation of nonlinear dynamics 10 models in the form of state space with state-dependent parameters (SDC-form)." it is not clear what SDC stands for. Please replace with state-dependent coeffient

The cooperation of robotic systems has many applications in the industry such as XX, XX, and XX. Fulfill the XXs.

However, there are other applications that require coordinated control, such as repair and welding work, space transportation of objects, and others. The "others" here does not povide much information.

Most of them do not exclude the possi bility of operating in an underwater environment -> exclude or include? What is the feature of under water? Why exclude underwater cases?

"One of the strongest research trends regards the use of Underwater Vehicle Manip-35 ulator Systems (UVMS) for the execution of intervention tasks, i.e. the tasks that require 36 the manipulation of objects or interaction with the environment." This paragraph is not well connected to its preceding and following paragraphs.

Most underwater manipulator tasks can be performed more efficiently when multi-38 ple manipulators are used cooperatively. -> Provide references. Show how this sort of applications makes sense. This sentence does not have much information.

Author Response

Response to Reviewer 3 Comments

Point 1: In the abstract, when you present "Here we used the approach based on the representation of nonlinear dynamics 10 models in the form of state space with state-dependent parameters (SDC-form)." it is not clear what SDC stands for. Please replace with state-dependent coeffient.

 Response 1:  Corrected. The annotation now contains the correct transcript of the SDC-form.

Point 2: The cooperation of robotic systems has many applications in the industry such as XX, XX, and XX. Fulfill the XXs.

Response 2: Thanks for your nice suggestions. The word and sentence order has been corrected as noted.

Point 3: However, there are other applications that require coordinated control, such as repair and welding work, space transportation of objects, and others. The "others" here does not povide much information.

Response 3: Thank you for your precious comments.  The first section of the paper has been revised. The coordinated control appendices are referenced to the source. Moreover, an overview of the UVMS has been expanded with a brief description of functional features in control tasks.

Point 4: Most of them do not exclude the possibility of operating in an underwater environment -> exclude or include? What is the feature of underwater? Why exclude underwater cases?

Response 4: The "do not exclude the possibility" part means that manipulation robots are capable of working in both air and underwater environments.

Point 5: "One of the strongest research trends regards the use of Underwater Vehicle Manipulator Systems (UVMS) for the execution of intervention tasks, i.e. the tasks that require the manipulation of objects or interaction with the environment." This paragraph is not well connected to its preceding and following paragraphs.

Response 5: Thank you for your precious comments. In revising the first section of the paper, a number of sentences were added to connect in meaning and extend the understanding of the relevant paragraphs.

Point 6: Most underwater manipulator tasks can be performed more efficiently when multiple manipulators are used cooperatively. -> Provide references. Show how this sort of applications makes sense. This sentence does not have much information.

Response 6: Corrected. A reference to a source has been attached to the relevant sentence, which allows for a detailed consideration of the issue described.

Round 2

Reviewer 2 Report

The authors have addressed the reviewer comments.

Reviewer 3 Report

.